# How Workplace Social Capital Affects Turnover Intention: The Mediating Role of Job Satisfaction and Burnout

**DOI:** 10.3390/ijerph19159587

**Published:** 2022-08-04

**Authors:** Huan Zhang, Lin Sun, Qiujie Zhang

**Affiliations:** 1School of Social Development and Public Policy, Beijing Normal University, Beijing 100875, China; 2School of Social Welfare, Beijing Vocational College of Labor and Social Security, Beijing 100029, China

**Keywords:** social work, social capital, turnover intention, job burnout, job satisfaction

## Abstract

Committed social workers are significant to organizational performance and service quality; therefore, it is crucial to explore the contributing factors of turnover intention to enhance social workers’ commitment. To reduce social workers’ turnover intention, this study used the first national survey data (N = 5620) of social workers in China to find out the relationship between workplace social capital and turnover intention in public service and explore possible solutions. This study treated workplace social capital as a comprehensive measure that captured employees’ overall perceptions of their interpersonal relations in the public sector. It covered the impact of many other organizational factors on turnover intention, such as job embeddedness, social networks, social relations, communication, and organizational fairness. The results confirmed that workplace social capital had a significant negative impact on employees’ turnover intention. Workplace social capital could be a better predictor of employees’ turnover intention than a single organizational factor or a combination of several factors. These findings not only deepened the theoretical understanding of social capital within the organization and brought insight into how workplace social capital affected employees’ turnover but also promoted a formation of a holistic organizational perspective from the fragmented organizational factors. Results also showed that job burnout and job satisfaction mediated the relation between workplace social capital and turnover intention. Public service agencies should endeavor to foster an organizational climate of cooperation and trust, encourage teamwork and altruistic behaviors among coworkers to reduce emotional exhaustion, and strengthen the professional identity and professional value of social work.

## 1. Background

The challenges of social workers’ turnover are particularly acute in the public sector [1,2]. According to Ferrin [3], the employee turnover rate reached an average of 73.8% in the hospitality industry. It results in a series of negative impacts on the organizations, such as reduced organizational performance [4], disrupted outcomes [5], lowered product/service quality [6] and loss of organizational knowledge [7]. Employee turnover has attracted the attention of both scholars and practitioners for a long time [8]. Social capital within organizations or workplaces can provide an important mechanism for reducing employee turnover theoretically [9]. Vagharseyyedin et al. [10] found that workplace social capital had a positive effect on the affective organizational commitment of employees. Jutengren et al. [11] revealed a positive association between social capital and work engagement and job satisfaction. Social capital is used to understand embeddedness and connections in a network or community [12,13,14] and reflect the character of social relations within organizations [15]. Social capital within organizations or workplaces can increase employee commitment to the organization [15]. Social networks [2], job embeddedness [16], trust [17], “guanxi” [18], coworker support [19], and communication [20] are found to be related to employee turnover. Guanxi is a concept derived from China, referring to important personal relationships with key persons who can help achieve certain goals [21]. However, social capital has been largely neglected in employee turnover studies as a single concept or construct. This study seeks to examine how social capital in the workplace affects turnover intention in public service. The first contribution is to test the role of workplace social capital as a single concept on turnover intention. We hypothesize that high workplace social capital reduces turnover intention because employees who perceive a high workplace social capital will have more affective connections, psychological contracts, perceived obligations, and constituent commitments with their coworkers or groups within the organization [22]. The second contribution is to examine the mechanisms by which workplace social capital affects turnover intention. Social capital is the core factor of organizational conditions in the employee turnover decision process. However, the influence of social capital would first be reflected in individual psychological perception and then further affect decision-making intention. Job burnout and job satisfaction may be the main intermediate psychological perception and attitude [23] We hypothesize that job burnout and job satisfaction mediate the relationship between social capital and turnover intention.

## 2. Literature Review and Hypotheses

### 2.1. Turnover and Turnover Intention

Employee turnover is more of a process that includes a range of activities, from the consideration of leaving to actually leaving [24]. A high turnover rate results in a series of negative impacts on the organizations, such as reduced organizational performance [4] and lowered service quality [6], as well as direct and hidden economic costs [25]. Turnover intention refers to the willingness to leave the working organization after careful consideration [26]. It is the leading cause and the best single predictor of actual turnover behavior [27]. Actually, reducing employees’ turnover intention is more valuable for organizations in preventing turnover, even if turnover intention does not automatically lead to actual turnover [28]. This study focuses on turnover intention as the outcome variable.

### 2.2. Workplace Social Capital and Turnover Intention

Social capital is believed to affect individuals’ access to resources in the organization, thus affecting their voluntary turnover [14]. Bonding, bridging, and linking social capital are three sub-types of social capital. Bonding social capital refers to the relationships among individuals with similar social identities and values. Bridging social capital refers to connections across groups of different ages, classes, and networks. Linking social capital refers to the trusting relationships between individuals of different authority gradients or statuses [29]. These three sub-types of social capital concentrate in organizations or workplaces [30]. Leana and Van Buren [15] developed organizational social capital as a resource reflecting the character of social relations within the organization and suggested that organizational social capital was related to stability in employment relationships. Dess and Shaw [9] found that organizational social capital could affect employee turnover. Since then, more and more features of social capital have been used as organizational conditions to predict employee turnover. A social network is one of the core concepts of social capital [13]. Social networks within groups in organizations are associated with turnover [31]. Mossholder, Settoon, and Henagan [32] tested that several relational variables, such as network centrality and perceived coworker support, were related to turnover. Moynihan and Pandey [2] used an intra-organizational social network to represent the social relation between employees and their coworkers and empirically supported the significant impact of social networks on turnover intention. Coworker friendship networks also have a significant impact on turnover intention [33]. Job embeddedness theory has become a widely accepted framework for understanding employee turnover [8]. Numerous studies showed that job embeddedness was a good predictor of turnover intention [7,34,35,36,37]. Some other facets of social capital have also been found to affect employee turnover, such as trust [17], guanxi [18], coworker support [19], communication [20], and perceptions of social relationships in the workplace [38]. However, few studies use social capital as a single concept or construct to examine its impact on employee turnover.

The idea of organizational social capital can further focus on workplace social capital because employees’ social relations in organizations are more focused on their workplaces. All three elements of social capital, including bonding, bridging, and linking social capital, can be found in the workplace [30]. We propose the following hypothesis:

**H1**.
*Workplace social capital is negatively related to turnover intention.*


### 2.3. The Mediating Role of Job Satisfaction and Job Burnout

The impact of workplace social capital on turnover intention could be mediated through employees’ job attitudes and professional perceptions [1,8]. Job satisfaction is a simple single summary measure of employees’ job attitudes [2]. It is the central factor in the traditional turnover model [8]. Many empirical studies have confirmed a consistent and significant negative relationship between job satisfaction and turnover [35,39]. 

Generally, workplace social capital brings more benefits to employees, such as increasing the associability and trust with their coworkers and supervisors and then increasing employees’ job satisfaction [15,40]. Social capital at work also helps form a sustainable work environment that would improve employees’ job satisfaction [41,42,43]. To test the above mechanism, we propose the following hypothesis:

**H2-1**.
*Job satisfaction mediates the negative relationship between workplace social capital and turnover intention.*


Job burnout is considered another important predictor of turnover intention in meta-analysis studies [44,45] Job burnout includes three key dimensions, “overwhelming exhaustion, feelings of cynicism and detachment from the job, and a sense of ineffectiveness and lack of accomplishment” [46]. Exhaustion is associated with job withdrawal, including turnover intention and actual turnover. Cynicism and ineffectiveness at work are associated with decreased job satisfaction and organizational commitment, which are both related to turnover intention [47]. Conservation of resources theory suggests that burnout reduces employees’ resources for a sustainable career and leads to dissatisfaction and low self-efficacy and eventually results in turnover [48,49]. Job burnout is a psychological characteristic or symptom in the workplace [46,47]. In meta-analysis studies, burnout is negatively related to job satisfaction [44].

Workplace social capital provides employees with more social resources in an organization [50]. High social capital in the workplace represents good relations, trust, and commitment among coworkers and contributes to the quality of life of employees at work [51]. Job burnout is often considered the opposite of career success and high quality of life (Barthauer et al., 2000). Many empirical studies supported that social capital in the workplace is negatively associated with job burnout [19,20,52,53,54]. However, no empirical studies directly measured workplace social capital but used some features of social capital to replace social capital. Workplace social capital as a single latent variable would be more theoretically valuable. We therefore propose the following hypotheses:

**H2-2**.
*Job burnout mediates the negative relationship between workplace social capital and turnover intention.*


**H2-3**.
*Job burnout mediates the positive relationship between workplace social capital and job satisfaction.*


### 2.4. Demographic Factors and Income

Previous studies point out the significant impact of demographic factors on turnover intentions [1,2]. However, it is difficult for demographic factors to explain the mechanism for turnover intention. Therefore, this study used demographic factors as control variables. Low salary leads to turnover intention [55]. Employees often judge their salary based on comparisons with coworkers or their expectations [23]. This results in the fact that salary is not as significant as other predicting factors of turnover [56]. Therefore, this study used salary as a control variable.

## 3. Methods

### 3.1. Participants and Procedures

This study used data from the Chinese Social Workers Survey (CSWS) conducted at the end of 2018. The CSWS is the first nationally representative survey of social workers in China. The sample size is 5620, which is about 0.55% of the total 1,025,757 social workers in China in 2017 [57]. In each province, we recruited one to four key persons who were well known in the province’s social work field. The key persons were selected based on one of the following criteria: (1) be a government official directly in charge of social work in the province; (2) be a principal staff of the provincial Association of Social Work/Federation of Social Workers; (3) be the principal leader of a widely influential social work organization in the province; or (4) be a social work professor at a local university who is active in the field of social work practice in the province. These selected key persons in each province sent the network link of the questionnaire to the social workers in their provinces.

An online questionnaire was completed by respondents. The procedure followed the recommended methods for minimizing the risk of common method variance [58]. Ethical approval was obtained in June 2018 from the Human Research Ethics Committee of the Authors’ institution. Of the valid respondents, a majority were female (77.6%), and their average age was 32.49 (SD = 8.05). Their average annual salary after tax was USD 6998 (SD = 4223). In total, 68.2% of them were certified social workers, and 40.8% had a master/bachelor’s degree in social work. Almost half of the respondents worked in the public sector (44.0%), and the rest worked in social sector not funded by the government (56.0%). Table 1 lists more details of the respondents.

### 3.2. Measures

Turnover intention (TI). Turnover intention includes not only the idea of wanting to leave but also the full process before actually leaving. Therefore, we used five items to measure TI. Three items were from Auerbach et al.’s scale [27]: thoughts of leaving, the feeling of having no future, and looking for information. Two items were from Jiang et al. [36]: “leaving this organization” and “leaving this profession”. All items used a five-point Likert response scale (1 = strongly disagree; 5 = strongly agree, Cronbach’s α = 0.916).

Workplace social capital (WSC). Workplace social capital was measured with the Chinese validated and tested version of an eight-item measure of social capital [30,59]. WSC is divided into three subscales: three items on bonding social capital (e.g., “We have a ‘we are together’ attitude”), two items on bridging social capital (e.g., “People in the workplace cooperate in order to help develop and apply new ideas”), and three items on linking social capital (e.g., “We can trust our supervisor”). All the items used a five-point Likert response scale (1 = strongly disagree; 5 = strongly agree; Cronbach’s α = from 0.916 to 0.927).

Job Burnout (JB). Job burnout was measured with the Chinese social work version of the Maslach Burnout Inventory–General Survey (MBI-GS) [60]. The MBI-GS consists of three subscales: five items measuring exhaustion (e.g., “I feel used up at the end of the workday”), five items measuring cynicism (e.g., “I have become less enthusiastic about my work”), and six items measuring professional efficacy (e.g., “In my opinion, I am good at my job”). All the items are scored on a seven-point frequency rating scale ranging from 0 (=never) to 6 (=daily; Cronbach’s α = from 0.893 to 0.922).

Job satisfaction (JS) was measured with the Chinese validated and tested version from Jiang et al. [36]. Job satisfaction was measured with four items, including satisfaction with (a) being a social worker, (b) the organization I am working for, (c) work tasks and duties, and (d) the achievement of helping clients solve problems. All the items used a five-point Likert response scale (1 = strongly disagree; 5 = strongly agree; Cronbach’s α = 0.898).

### 3.3. Data Analysis

Confirmatory factor analysis was used to examine the construct validity of the four latent variables in AMOS 24. All hypotheses in the conceptual model were executed using linear regression analyses in SPSS 24 and PROCESS 2.16 [61]. Age, gender, Ln(salary), years of the participant as a social worker (SW years), whether the participant is working in the social sector (Employer type), whether the participant is a certified social worker (Certified), and whether the participant has a master/bachelor’s degree in social work (MSW/BSW) were added as control variables. Considering the possible nonlinear effects, the square of age and the square of SW years were also added as control variables. The mediation effect in the conceptual model was evaluated by bootstrapping because bootstrapping has higher power and controls Type I errors [61]. As recommended, we used the 95% confidence intervals (bias-corrected) and 5000 bootstrap samples with PROCESS 2.16 for SPSS [61].

## 4. Results

### 4.1. Descriptive Statistics and Bivariate Correlations

Confirmatory factor analysis was conducted to examine the construct validity of the studied variables. The results showed that the measurement model fit the data well (χ2/df = 13.50, SRMR = 0.042, CFI = 0.969, RFI = 0.961, RMSEA = 0.047). Table 2 presents the psychometric properties and correlations among the latent variables. The skewness and kurtosis of the variables ranged from −0.576 to 0.175, which conformed to the normal distribution. The results also showed that the composite reliability (CR) was good (>0.7), and the average variance extracted (AVE) indicated acceptable discriminant validity (>0.7) for all latent variables.

### 4.2. Regression Analyses

Table 3 presents the results from the regression analyses. Model 1 only included the control variables for the purposes of quantifying their influence on TI. Almost all the control variables influenced social workers’ TI, and the direction of influence was basically the same as the model expected. However, all the control variables only explained a trivial proportion of the total variance (R^2^ = 0.065). This suggested that demographic variables and salary were not the core factors affecting TI. Model 2 added WSC as an independent variable to Model 1 to test H1. The results showed that WSC negatively influenced TI (β = −0.522, *p* < 0.001) and explained 20.8% of the variance. This outcome supported H1. In Model 3, the mediators (JB) were regressed on the independent variable (WSC), and significant relationships were found. In Model 4, the mediators (JS) were regressed on the independent variables (WSC and JB), and significant relationships were also found. Model 5 indicated that both JB and JS were related to TI (β = 0.548, *p* < 0.001, β = 0.177, *p* < 0.001, respectively), while WSC was still related to TI; however, the absolute value of the coefficient was greatly reduced (from Model 2 (β = −0.522, *p* < 0.001) to Model 5 (β = −0.128, *p* < 0.001)). The inclusion of JB and JS in the model explained 24.9% of the variance for TI (from Model 2 (R^2^ = 0.208) to Model 5 (R^2^ = 0.457)). Therefore, JB and JS might be double-mediators in the relationship between WSC and TI.

### 4.3. Mediation Effect Analyses

PROCESS 2.16 for SPSS was used to test the mediation effect (Hayes, 2013). According to H2-1, H2-2, and H2-3, we used Model 6 (2 mediators) in PROCESS 2.16. We used the 95% confidence intervals (bias-corrected) and 5000 bootstrap samples.

Table 4 shows the results of the mediation effect analyses. Both the direct effect (95% confidence interval [−0.162, −0.092]) and the indirect effect (95% confidence interval [−0.428, −0.363]) were significant. Accordingly, the results supported H2-1, H2-2, and H2-3, which predicted that JB and JS partially double-mediated the relationship between WSC and TI. Comparatively, burnout had a larger indirect effect (72.08% of the total indirect effect).

Figure 1 shows the results of the double-mediator model formulated.

## 5. Discussion

The first contribution is to verify the significant impact of workplace social capital on employees’ turnover intention. Social capital is one of the most popular and widely used concepts about interpersonal relationships [62] Social capital in the workplace was also developed to understand the interpersonal relationships among employees [15,30]. Many studies have discussed the relationship between social capital and employee turnover based on resource dependence theory [15], human capital theory [9], job embeddedness theory [8,63], and organizational support theory [2]. However, these studies tested the impact of one or several facets of social capital on turnover and might miss some important information. This study treats workplace social capital as a comprehensive measure that captures employees’ overall perceptions of their interpersonal relations in the public sector. It covers the impact of many other organizational factors on turnover intention, such as job embeddedness, social networks, social relations, communication, and organizational fairness. The results confirmed that workplace social capital had a significant negative impact on employees’ turnover intention with an adequate explanation (ΔR^2^ = 14.3%). Workplace social capital could be a better predictor of employees’ turnover intention than a single organizational factor or a combination of several factors. These findings not only deepened the theoretical understanding of social capital within the organization and brought insight into how workplace social capital affected employees’ turnover but also promoted a formation of a holistic organizational perspective from the fragmented organizational factors. Li et al. [64] also measured social capital in a comprehensive way by testing the effect of three types of capital and their combined effect on hotel employees’ turnover intention.

The second contribution is the establishment of a mechanism for a deep understanding of how workplace social capital affects employees’ turnover intention. The results confirmed a partial double-mediating mechanism of job burnout and job satisfaction in workplace social capital that affects turnover intention. Job satisfaction plays a partial mediating role between workplace social capital and turnover intention. Job satisfaction is considered one of the most important employee job attitudes affecting turnover intention [8,65]. Workplace social capital reflects the interpersonal relationships in the organization and constitutes a basic and important organizational condition for employees’ satisfaction with their job. Therefore, it is natural that workplace social capital affects employees’ turnover intention through their job satisfaction. The mediation role of job satisfaction between work arrangement and turnover was also found by Berber et al. [66].

Job burnout also plays a partial mediating role between workplace social capital and turnover intention. Job burnout is an important concept describing employees’ psychological characteristics or symptoms associated with their job [46]. Job burnout is also a psychological factor affecting turnover intention [44,45]. Results showed that workplace social capital had a significant negative impact on employees’ turnover intention through their job burnout level. The findings were consistent with previous studies [19,67] and showed that strong social capital in the workplace was a tool against job burnout [63]. Third, job burnout and job satisfaction partially double-mediate the impact of workplace social capital on turnover intention. There is a significant negative relationship between job burnout and job satisfaction [44]. Generally, the employee’s psychological characteristics of his/her job change at first and then affect his/her job attitude. Job burnout is considered an antecedent and effective predictor of job satisfaction [44]. Therefore, there is an impact path from workplace social capital to job burnout, job satisfaction, and turnover intention. Besides the double-mediation of job burnout and job satisfaction, workplace social capital still has a significant negative impact on employees’ turnover intention. Last but not least, this study verified that salary and some demographic factors had a significant impact on turnover intention, although the effect of these factors was small. Most of these impacts are consistent with previous studies. Results showed that salary has a significant negative impact on turnover intention. Low salary is always one of the strongest predictors of employees’ turnover intention [8]. Results showed that males are positively related to turnover intention. This is in line with previous studies in both China and the U.S. [68]. Age was negatively related to turnover intention, and age squared was positively related to turnover intention. This means that the relationship between age and turnover intention was a U-shaped curve, which was consistent with previous studies of employee turnover in the public sector [69,70]. However, the U-shaped curve between age and social workers’ turnover intention is a new finding in China. Generally, younger social workers are more likely to turnover because they may have more job alternatives and more flexibility [1,36]. However, due to the rapid development of the social work service in China, especially the rapid growth of social work service organizations [71], some older social workers with rich experience in public services have become more popular and have more job alternatives; however, older social workers unable to continue to undertake heavy work burden are also considering turnover. These dual reasons lead to an increase in the older social workers’ turnover intention. It is more interesting that professional tenure is positively related to turnover intention and that tenure squared is negatively related to turnover intention in this study. This means that the relationship between tenure and turnover intention is an inverted U-shaped curve. These results are new findings in the study of Chinese social workers’ turnover intention, which are inconsistent with existing studies in the Western context [1,72]. Combining the results of the above two U-shaped curves further suggests that the above dual reasons explain the relationship between age, tenure, and turnover intention of social workers in current China. The results also showed that social workers with MSW or BSW degrees and certificates are positively related to turnover intention. Evidently, social workers with higher professional competence are more likely to turnover because they have more job opportunities. Other studies have shown similar impact relationships [1,73]. It should be pointed out that demographic factors have only weak relationships with turnover intention, although many of them have a statistically significant impact on turnover intention.

### 5.1. Practical Implications

Results suggest two additional ways to reduce employees’ turnover intention. One way is to improve employees’ job satisfaction. Job satisfaction is the most direct attitude affecting turnover intention [2,8]. Some features of workplace social capital, such as cooperation and trust, are particularly important for improving employees’ job satisfaction. Therefore, public service agencies should pay attention to fostering a good organizational climate of cooperation and trust. The other way is to reduce employees’ job burnout. Burnout is one of the most frequently studied phenomena in the working population. Burnout is also an important challenge in the careers of social workers [60,74,75]. Two of the three dimensions of burnout, namely, emotional exhaustion and cynicism, are more closely related to turnover intention [60]. Encouraging teamwork and altruistic behaviors among coworkers is helpful in reducing emotional exhaustion. Strengthening the professional identity and professional value of social work is also useful in solving the problem of cynicism. Professional supervision improves workplace social capital and helps reduce the burnout of social workers meanwhile [76].

### 5.2. Limitations and Recommendations for Future Research

This study has several limitations. First, the cross-sectional design does not allow the assessment of the theoretical model variables over time or of causal inferences [77]. Future research can use longitudinal tracking data to provide more causal linkages between workplace social capital and turnover. Second, this study only discusses turnover intention, not actual turnover behavior. Actual turnover may be more interesting for theoretical and practical implications. Future research should continue to examine the relationships between workplace social capital and actual turnover.

## 6. Conclusions

This article explored the affect of workplace social capital and turnover intention in public service to find the contributing factors of turnover intention to enhance social workers’ commitment. This study treated workplace social capital as a comprehensive measure that captured employees’ overall perceptions of their interpersonal relations in the public sector. It covered the impact of many other organizational factors on turnover intention. Results showed that workplace social capital had a significant negative affect on social workers’ turnover intention. Workplace social capital could be a better predictor of employees’ turnover intention than a single organizational factor or a combination of several factors. These findings not only deepened the theoretical understanding of social capital within the organization and brought insight into how workplace social capital affected employees’ turnover but also promoted a formation of a holistic organizational perspective from the fragmented organizational factors. Results also revealed that job burnout and job satisfaction played a mediation role between workplace social capital and turnover intention. Public service agencies should endeavor to foster an organizational climate of cooperation and trust, encourage teamwork and altruistic behaviors among coworkers to reduce emotional exhaustion, and strengthen the professional identity and professional value of social work.

## Figures and Tables

**Figure 1 ijerph-19-09587-f001:**
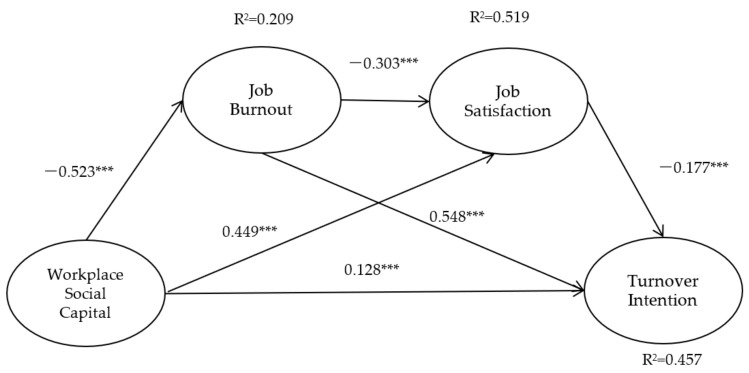
The results of double-mediator model. Note. *** *p* < 0.001.

**Table 1 ijerph-19-09587-t001:** Participant Demographics.

Variables	Mean	*SD*	Variables		No. (%)
Age	32.49	8.05	Gender	Male	1259 (22.4%)
SW years ^a^	5.14	4.93		Female	4361 (77.6%)
Income ^b^	7.09	4.11	Employer type	Social sector	3145 (56.0%)
Variables		No. (%)		Public sector/community	2475 (44.0%)
Education level	High school and below	220 (3.9%)	Certified ^c^	Yes	3832 (68.2%)
	College	1554 (27.7%)		No	1788 (31.8%)
	Undergraduate	3321 (59.1%)	MSW/BSW ^d^	Yes	2295 (40.8%)
	Graduate and above	525 (9.4%)		No	3325 (59.2%)

Note: N = 5620. ^a^ Years of the participant as social worker; ^b^ Participant’s work income after tax in 2017 ($1000); ^c^ Is the participant a certified social worker? ^d^ Does the participant have a master/bachelor’s degree in social work?

**Table 2 ijerph-19-09587-t002:** Psychometric properties and correlations among the latent variables.

	Mean	SD	Skewness	Kurtosis	CR	AVE	1	2	3	4
1. TI	2.749	1.032	0.076	−0.576	0.913	0.679	0.820			
2. WSC	3.977	0.752	−0.444	0.176	0.930	0.817	−0.403 **	0.904		
3. JB	2.332	0.947	0.100	−0.086	0.730	0.533	0.646 **	−0.433 **	0.730	
4. JS	3.804	0.751	−0.262	0.161	0.887	0.665	−0.503 **	0.624 **	−0.584 **	0.815

Note: (1) N = 5620. (2) TI—turnover intention; WSC—workplace social capital; JB—job burnout; JS—job satisfaction. (3) CR—composite reliability; AVE—average variance extracted. (4) Good: CR > AVE, AVE > 0.5, CR > 0.7. (5) The underlined figures are the square roots of the AVE. (6) ** *p* < 0.01 (2-tailed).

**Table 3 ijerph-19-09587-t003:** Regression analyses.

	Model 1	Model 2	Model 3	Model 4	Model 5
Variable	TI	TI	JB	JS	TI
WSC		−0.522 ***	−0.523 ***	0.449 ***	−0.128 ***
JB				−0.303 ***	0.548 ***
JS					−0.177 ***
Ln(salary)	−0.177 ***	−0.167 ***	−0.083 **	0.016	−0.114 ***
Gender (female = 1)	−0.148 ***	−0.132 ***	−0.064 *	−0.047 **	−0.101 ***
Age	−0.072 ***	−0.066 ***	−0.055 ***	0.007	−0.032 **
Age squared	0.001 **	0.001 **	0.001 **	0.0000	0.0003 †
SW years	0.035 ***	0.039 ***	0.037 ***	0.008 †	0.018 **
SW years squared	−0.001 **	−0.001 ***	−0.001 ***	−0.0003 †	−0.001 **
MSW/BSW	0.077 *	0.062 *	0.055 *	0.043 *	0.036
Employer type	−0.132 ***	−0.109 ***	−0.090 ***	0.028 †	−0.050 *
Certified	0.171 ***	0.101 **	0.060 *	−0.026	0.061 *
Constant	6.130	7.983	6.474	2.397	4.513
R2	0.065	0.208	0.209	0.519	0.457
F	30.824	107.536	108.225	409.926	297.746
DW	1.976	1.989	1.956	2.008	1.997

Note: (1) N = 5331. (2) † *p* < 0.1; * *p* < 0.05; ** *p* < 0.01; *** *p* < 0.001. (3) TI—turnover intention; WSC—workplace social capital; JB—job burnout; JS—job satisfaction. (4) SW years = Years of the participant as social worker; MSW/BSW = Did the participant have a master/bachelor’s degree in social work; Employer type = Did the participant work in social sector (=1) or public sector (=0); Certified = Was the participant certified social worker?

**Table 4 ijerph-19-09587-t004:** Mediation effect.

	Effect	SE (Boot SE)	t	*p*	95% CI (Bias Corrected)	
					Lower	Upper
Total	−0.521	0.017	−30.914	***	−0.554	−0488
Direct	−0.127	0.018	−7.113	***	−0.162	−0.092
Indirect	−0.394	0.017		***	−0.428	−0.363
Job burnout	−0.287	0.012		***	−0.312	−0.264
Job burnout -> Job satisfaction	−0.028	0.004		***	−0.036	−0.020
Job satisfaction	−0.079	0.011		***	−0.100	−0.058

Note: (1) N = 5331. (2) *** *p* < 0.001. (3) CI—confidence intervals.

## Data Availability

Not available.

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
