# Peer review of "How Workplace Social Capital Affects Turnover Intention: The Mediating Role of Job Satisfaction and Burnout"

_ijerph, 2022, doi:10.3390/ijerph19159587_

Round 1
Reviewer 1 Report
Dear author(s),
I find your paper well-written and well-presented. The methodology is based on standardized questionnaires. Results are clear.
However, I would like to suggest some additional issues to be solved in order to improve your results.
- Please, state the aim of your research in the abstract.
- Provide the source for the Job Satisfaction questionnaire.
- Explain all constructs with more sources – turnover, turnover intentions, job burnout, job satisfaction and Workplace social capital.
- Add additional sources into your research, that are in the line with the research:
- Vagharseyyedin, S. A., Zarei, B., & Hosseini, M. (2018). The role of workplace social capital, compassion satisfaction and secondary traumatic stress in affective organisational commitment of a sample of Iranian nurses. Journal of Research in Nursing, 23(5), 446-456.
- Berber, N., Gašić, D., Katić, I., & Borocki, J. (2022). The Mediating Role of Job Satisfaction in the Relationship between FWAs and Turnover Intentions. Sustainability, 14(8), 4502.
- Jutengren, G., Jaldestad, E., Dellve, L., & Eriksson, A. (2020). The potential importance of social capital and job crafting for work engagement and job satisfaction among health-care employees. International Journal of Environmental Research and Public Health, 17(12), 4272.
- Li, Z., Yu, Z., Huang, S. S., Zhou, J., Yu, M., & Gu, R. (2021). The effects of psychological capital, social capital, and human capital on hotel employees’ occupational stress and turnover intention. International Journal of Hospitality Management, 98, 103046.
Author Response
Dear Professor,
Thank you so much for your kindly comments! They are very enlightening for me and my team workers. We have made rivisions according to your advice as follows:
- Please, state the aim of your research in the abstract.
Re: We added the aim of our research in the abstract.
2. Provide the source for the Job Satisfaction questionnaire.
Re: We provided the source for the Job Satisfaction questionnaire.
3.Explain all constructs with more sources – turnover, turnover intentions, job burnout, job satisfaction and Workplace social capital.
Re: We tried hard to explain some of the constructs. If it is not enough, please let us know.
4.Add additional sources into your research, that are in the line with the research:
Vagharseyyedin, S. A., Zarei, B., & Hosseini, M. (2018). The role of workplace social capital, compassion satisfaction and secondary traumatic stress in affective organisational commitment of a sample of Iranian nurses. Journal of Research in Nursing, 23(5), 446-456.
Berber, N., Gašić, D., Katić, I., & Borocki, J. (2022). The Mediating Role of Job Satisfaction in the Relationship between FWAs and Turnover Intentions. Sustainability, 14(8), 4502.
Jutengren, G., Jaldestad, E., Dellve, L., & Eriksson, A. (2020). The potential importance of social capital and job crafting for work engagement and job satisfaction among health-care employees. International Journal of Environmental Research and Public Health, 17(12), 4272.
Li, Z., Yu, Z., Huang, S. S., Zhou, J., Yu, M., & Gu, R. (2021). The effects of psychological capital, social capital, and human capital on hotel employees’ occupational stress and turnover intention. International Journal of Hospitality Management, 98, 103046.
Re: Thank you for your referrence to these articles! They are so important to our research. We have gained much more insight into this field.
In all, your patience and kindly help is highly appreciated by all of us!
Reviewer 2 Report
A well-written and well-organized manuscript. I agree with the authors' argument on how important social capital is to workplace turnover and job satisfaction. The methodology is sound and the results are well presented. However, I have some comments to make the manuscript more appeal to a wider readership:
(i) Elaborate more on the importance of social capital (in lay terms) and give examples of related concepts in the Literature review. While the authors did cited studies that supported the relationship between social capital and turnover rate. They can add more details such as statistical results from those studies to better positioning their own findings later in the manuscript. Also, spelling them out all the concepts will be appreciated by readers who are not familiar with the topic. For example, "guanxi" - personal connections.
(ii) Discuss recent studies on social capital and turnover rate such as Li et al. (2021) - https://www.sciencedirect.com/science/article/pii/S0278431921001894 and Andresen,Goldmann,& Volodina (2018) - https://onlinelibrary.wiley.com/doi/full/10.1111/emre.12120
(iii) Provide specific statistics (e.g., revenue lost) on why turnover in social workers are critical to study.
(iv) Describe more of the data collection process (e.g., how did the authors gained access to the sample, did they send reminders to participants, etc.).
(v) Present some of the descriptive statistics (e.g., in a table) to help the general audience get an overview of the results.
Author Response
Dear Professor,
Thank you for your kindly comments for our study! It is so enlightening to us! We have made revisions according to your advice as follows:
(i) Elaborate more on the importance of social capital (in lay terms) and give examples of related concepts in the Literature review. While the authors did cited studies that supported the relationship between social capital and turnover rate. They can add more details such as statistical results from those studies to better positioning their own findings later in the manuscript. Also, spelling them out all the concepts will be appreciated by readers who are not familiar with the topic. For example, "guanxi" - personal connections.
Re: We added more explanations on the importance of social capital and results from previous studies which supported the relationship between social capital and turnover. We explained the concepts "guanxi" in this article.
(ii) Discuss recent studies on social capital and turnover rate such as Li et al. (2021) - https://www.sciencedirect.com/science/article/pii/S0278431921001894 and Andresen,Goldmann,& Volodina (2018) - https://onlinelibrary.wiley.com/doi/full/10.1111/emre.12120
Re: This article is important to our study. Thank you for your referrence to us. We have added the findings from this article.
(iii) Provide specific statistics (e.g., revenue lost) on why turnover in social workers are critical to study.
Re: We added the high turnover rate in hospitality industry and more negative results from turnover. However, we did not find specific statistics on training cost or revenue lost. We guess it is beacuse these costs are hidden and not easy for calculation.
(iv) Describe more of the data collection process (e.g., how did the authors gained access to the sample, did they send reminders to participants, etc.).
Re: We added the data collection process.
(v) Present some of the descriptive statistics (e.g., in a table) to help the general audience get an overview of the results.
Re: We presented a table for the descriptive statistics. Please see Table 1.
In all, your patience and kindly help is highly appreciated by us!